# Exploring the Gelation Mechanisms and Cytocompatibility of Gold (III)-Mediated Regenerated and Thiolated Silk Fibroin Hydrogels

**DOI:** 10.3390/biom10030466

**Published:** 2020-03-18

**Authors:** Chavee Laomeephol, Helena Ferreira, Supansa Yodmuang, Rui L. Reis, Siriporn Damrongsakkul, Nuno M. Neves

**Affiliations:** 1Biomedical Engineering Research Center, Faculty of Engineering, Chulalongkorn University, Bangkok 10330, Thailand; chavee.l@student.chula.ac.th (C.L.); Supansa.Y@chula.ac.th (S.Y.); 2Biomaterial Engineering for Medical and Health Research Unit, Faculty of Engineering, Chulalongkorn University, Bangkok 10330, Thailand; 33B’s Research Group, I3Bs-Research Institute on Biomaterials, Biodegradables and Biomimetics, University of Minho, Headquarters of the European Institute of Excellence on Tissue Engineering and Regenerative Medicine, AvePark-Parque de Ciência e Tecnologia, Zona Industrial da Gandra, 4805-017 Barco, Guimarães, Portugal; helenaferreira@i3bs.uminho.pt (H.F.); rgreis@i3bs.uminho.pt (R.L.R.); 4ICVS/3B’s-PT Government Associate Laboratory, 4806-909 Braga/Guimarães, Portugal; 5Research Affairs, Faculty of Medicine, Chulalongkorn University, Bangkok 10330, Thailand; 6The Discoveries Centre for Regenerative and Precision Medicine, Headquarters at University of Minho, Avepark, 4805-017 Barco, Guimarães, Portugal; 7Department of Chemical Engineering, Faculty of Engineering, Chulalongkorn University, Bangkok 10330, Thailand

**Keywords:** silk fibroin, thiolated silk fibroin, gold, hydrogel, cytocompatibility

## Abstract

Accelerating the gelation of silk fibroin (SF) solution from several days or weeks to minutes or few hours is critical for several applications (e.g., cell encapsulation, bio-ink for 3D printing, and injectable controlled release). In this study, the rapid gelation of SF induced by a gold salt (Au^3+^) as well as the cytocompatibility of Au^3+^-mediated SF hydrogels are reported. The gelation behaviors and mechanisms of regenerated SF and thiolated SF (tSF) were compared. Hydrogels can be obtained immediately after mixing or within three days depending on the types of silk proteins used and amount of Au^3+^. Au^3+^-mediated SF and tSF hydrogels showed different color appearances. The color of Au-SF hydrogels was purple-red, whereas the Au-tSF hydrogels maintained their initial solution color, indicating different gelation mechanisms. The reduction of Au^3+^ by amino groups and further reduction to Au by tyrosine present in SF, resulting in a dityrosine bonding and Au nanoparticles (NPs) production, are proposed as underlying mechanisms of Au-SF gel formation. Thiol groups of the tSF reduced Au^3+^ to Au^+^ and formed a disulfide bond, before a formation of Au^+^-S bonds. Protons generated during the reactions between Au^3+^ and SF or tSF led to a decrease of the local pH, which affected the chain aggregation of the SF, and induced the conformational transition of SF protein to beta sheet. The cytocompatibility of the Au-SF and tSF hydrogels was demonstrated by culturing with a L929 cell line, indicating that the developed hydrogels can be promising 3D matrices for different biomedical applications.

## 1. Introduction

Silk fibroin (SF) is a protein-based biopolymer derived from *Bombyx mori* mulberry silkworms. SF is the main component presented in silkworm cocoons (about 72–81%) and another component is a glue-like sericin protein that can be eliminated by alkaline treatments. SF is a heterodimeric protein, containing a heavy chain (MW~325 kDa) and a light chain (MW~25 kDa), linked by disulfide bridges. The glycine-alanine repetitive sequences, present in high amount in the heavy chain, are responsible for the formation of beta sheet structure, making SF a natural-derived material with good biodegradability and excellent mechanical properties [1]. Due to its biocompatibility, the high mechanical stability and resistance against several physiological enzymes, SF has a long history of use in biomedical applications, e.g., suture materials [2].

SF fibers obtained from silk cocoons can be readily used, but they have to be dissolved in chaotropic agents to disrupt the hydrogen bonds. By doing this, the ordered beta sheet structures turn into amorphous random coil, and the dissolution of SF fiber is obtained. The SF solution can be further processed into membranes, microparticles, fibers, or porous scaffolds, depending on the specific application envisaged [2]. However, SF solution spontaneously turns into hydrogel within several days or weeks because of the gradual transition to the thermodynamically stable beta sheet structure [3]. This self-assembly characteristic of SF has intrigued many researchers, which have been employing various strategies to control and accelerate the gelation process. Indeed, a short gelation time is crucial for many applications, such as cell-loaded hydrogels or injectable controlled release systems. Crosslinking agents (e.g., glutaraldehyde, carbodiimides, or genipin), enzymes (e.g., tyrosinase or peroxidase), physical methods (e.g., vortexing or sonication), or chemical additives (e.g., alcohols [4], surfactants [5], or phospholipids [6]) have been applied to control the kinetics of SF gelation [7].

Gold (Au) is a transition metal that has been widely used for medical purposes, namely as a chrysotherapy for rheumatoid arthritis, or as a chemotherapeutic agent. Au salts are also used to produce Au nanoparticles (AuNPs) that display a high potential in photodynamic and photothermal therapies for cancers, due to their physical and optical properties [8]. Furthermore, AuNPs are used as contrast agents for imaging, carriers for macromolecules, and biosensors [9,10]. Several chemical and physical methods are reported for the production of AuNPs, including those using plant extracts or natural proteins [9,11]. In the presence of proteins, Au salts (the most stable form is Au^3+^) are reduced by amino (-NH_2_) or thiol (-SH) groups to Au^+^. Afterwards, tyrosine, of which its cresol group displays a strong electron donating property, reduces Au^+^ to Au, and tyrosyl radical is converted to dityrosine [12]. However, when thiol groups are presented, the reduction to Au is prevented due to strong bond formation between Au^+^ and sulfur [13,14].

Due to the presence of amino group containing residues as well as a high content of tyrosine in SF (~5.75%) [15], it can be used in the production of AuNPs by a simple procedure and without hazardous reagents [11,16]. Moreover, the preparation of hydrogels using thiolated polymers (e.g., thiolated polyethylene glycol) crosslinked by Au^3+^ was previously reported [17,18].

In this work, we developed SF hydrogel systems using Au^3+^ salt as a chemical gelator. Both SF and thiol-functionalized SF (tSF) were selected as base matrices to investigate and compare their mechanisms of gel formation in the presence of Au^3+^. Furthermore, we evaluated the cytocompatibility of the obtained hydrogels to assess their potential for biomedical applications.

## 2. Materials and Methods

### 2.1. Materials

“Nangnoi Srisaket 1” *Bombyx mori* Thai silk cocoons were kindly provided by Queen Sirikit sericulture center, Srisaket province, Thailand. All chemical reagents were of analytical grade and purchased from Sigma-Aldrich, St. Louis, MO, USA, unless otherwise stated. Culture media and reagents used for biological experiments were supplied from Thermo Fischer Scientific, Waltham, MA, USA.

### 2.2. Preparation of SF and tSF Solutions

Silk cocoons were cut in half, boiled in 0.02 M Na_2_CO_3_ for 20 min, and washed with deionized (DI) water to remove silk glue. The SF fibers were then dissolved in 9.3 M LiBr at 1:4 ratio (*w*:*v*) and incubated at 60 °C for 4 h. Subsequently, the obtained amber solution was dialyzed against DI water using a dialysis tube with 12-16 kDa MWCO (Sekisui, Osaka, Japan) for 48 h. Finally, the solution was centrifuged at 4 °C and 9000 rpm for 20 min to remove debris. The concentration was determined from its dry solid weight. The SF solution was kept at 4 °C until use.

The introduction of sulfhydryl groups (-SH) into SF was performed according to Monteiro et al. [19] with slight modifications. SF solution of 6% *w/v* was dialyzed against 0.1 M sodium phosphate buffer (pH 8.0) at 4 °C overnight. Different amounts of 2-iminothiolane (2-IT) and 4-dimethylaminopyridine (DMAP) were mixed with the SF solution to obtain a final concentration ranging from 5 to 50 mM. Then, the solutions were gently stirred at 37 °C, prior to the dialysis with DI water and determination of the solution concentration by the dry solid weight method. For all further experiments, the tSF solution functionalized with 50 mM 2-IT and DMAP was used because the use of a higher amount of reagents was not possible due to solubility issues.

### 2.3. Quantification of Sulfhydryl Groups

The amount of sulfhydryl groups in tSF was analyzed by Ellman reagent assay. The 1% tSF solution was mixed with 0.1 M 5′-dithiobis-(2-nitrobenzoic acid) (DTNB) and 1 mM EDTA. The obtained mixtures were incubated at 37 °C during 3 h in the dark and the absorbance was measured at 412 nm. Cysteine was used as standard.

### 2.4. Gelation of SF and tSF with Au^3+^

SF and tSF solutions were mixed with gold (III) chloride trihydrate (HAuCl_4_.3H_2_0) to a final concentration of 3% silk protein and 0.5, 1, and 5 mM Au^3+^. The mixtures were incubated at 37 °C for 14 days and the changes were observed occasionally. To investigate the conformation transition, the lyophilized samples obtained from different time-points were analyzed using Fourier-transform infrared spectroscopy (FTIR) (IRPrestige 21, Shimadzu, Kyoto, Japan). The samples were ground with KBr and punched. The infrared spectra were collected between 4000 to 400 cm^−1^ in an absorbance mode with 2.0 cm^−1^ resolution and 1.0 cm^−1^ interval. The content of secondary structure was determined by Fourier self-deconvolution (FSD) and curve-fitting techniques according to Hu et al. [20]. The region between 1725 and 1575 cm^−1^ was firstly deconvoluted using Omnic 8.0 software (Thermo Fisher Scientific, Waltham, MA, USA). The spectral line shape was fitted using Voight function with a half-bandwidth of 10 cm^−1^ and an enhancement factor of 3.0. Afterwards, the deconvoluted spectrum was curve-fitted using Origin Pro 9.0 software (OriginLab, Northamptom, MA, USA). The beta sheet content was calculated from the sum of area under the peaks located in the 1616–1637 and 1696–1703 cm^−1^ regions. The amount of other structures, including tyrosine residue, random coil, alpha-helix, and beta turn, were determined from the area under the peaks located within 1595–1615, 1638–1655, 1656–1662, and 1663–1696 cm^−1^, respectively [20].

### 2.5. Determination of Bonds and AuNPs Formation in Au-SF or Au-tSF Hydrogels

The fluorescence emission characteristic of dissociated phenolic hydroxyl groups allows the assessment of the amount of dityrosine by fluorescence measurements [21]. Different amounts of Au^3+^, ranging from 0.1 to 5 mM, were added to 1% SF or 1% tSF solution. The pH of the solution was maintained at 7.4 using 10 mM HEPES buffer. Fluorescence measurements were performed using a spectrofluorometer (FP-8500, Jasco, Easton, MD, USA) at an excitation wavelength of 320 nm. The emission spectra between 350 and 500 nm were collected with 2-nm interval.

The availability of sulfhydryl groups after the addition of Au^3+^ was examined by the Ellman reagent assay as previously mentioned. Due to the low amount of SH groups present in SF, it was not possible to quantify them in the mixture of Au-SF. All experiments were performed in triplicate.

The formation of AuNPs was confirmed by determining surface plasmon resonance bands obtained from UV–vis absorbance spectra. SF and tSF solutions mixed with different Au^3+^ amounts were prepared and incubated at room temperature for 3 h in the dark. Absorption spectra between 300 and 700 nm were then collected using a microplate reader (Synergy HT, Bio-Tek, Winooski, VT, USA).

### 2.6. XPS Analysis

The hydrogel samples were freeze-dried, finely ground and stored in a desiccator prior to XPS analysis using an Axis ultra DLD spectrometer (Kratos Analytical, Kyoto, Japan) equipped with a monochromatic Al Kα x-ray source. Wide scan was performed at binding energy (BE) 0–1200 eV. The spectra of Au4f and S2p were collected in a region of 81–94 and 159–174 eV, respectively. Curve-fitting was performed using Origin Pro 9.0 software. The peaks in Au4f_7/2_ region were assigned into three peaks at approximately 84.0, 84.7, and 85.5 eV [22] and the peaks within S2p_3/2_ region were at 162.2 and 164.2 eV [23].

### 2.7. Micromorphological Assessment

Au-SF and Au-tSF hydrogels prepared with 1 mM Au^3+^ were freeze-dried, cut and sputter-coated with platinum. Surface feature analysis was visualized using a scanning electron microscope (SEM) (JSM-IT500HR, Jeol, Tokyo, Japan) at 300× and 15,000× magnifications.

### 2.8. Analysis of Viscoelastic Properties

The regenerated SF and tSF solutions and the mixtures of 3% SF or 3% tSF with 0.5 to 5 mM Au^3+^ were loaded in the sample holder of a Kinexus Pro+ rheometer (Malvern Instruments, Malvern, UK) equipped with parallel plate geometry with 1-mm gap. Samples were equilibrated at 37 °C for 10 min before measurement. Frequency sweep experiments were done and the moduli over 0.01–100 Hz were collected at a fixed strain of 0.1%. All experiments were done in triplicate.

### 2.9. Cell Culture

L929 mouse fibroblasts were kindly provided by Prof. Dr. Tanapat Palaga, Department of Microbiology, Faculty of Science, Chulalongkorn University. The cells were maintained in Dulbecco’s modified eagle medium (DMEM) containing 10% fetal bovine serum (FBS) and 1% penicillin/streptomycin. The subculture was performed when confluency reached approximately 80–90% using TrypLE Express to detach the adherent cells.

### 2.10. Cytocompatibility Evaluation

Solutions of 3% SF or tSF were sterilized by UV irradiation for 30 min and HAuCl_4_ solution was filtered through 0.22 µm syringe filter before use. SF and tSF gels were formed in 96-well plates by incubating the 3% solutions at 37 °C under a vapor-saturated atmosphere for 7 days. The Au-SF and Au-tSF hydrogels were prepared as previously described.

The hydrogel samples were washed with phosphate buffer saline (PBS), seeded with L929 fibroblasts (10,000 cells/cm^2^) and incubated in a humidified 5% CO_2_ atmosphere at 37 °C. Medium was changed every other day.

Cell metabolic activity was determined using the MTS assay (Abcam, Cambridge, MA, USA). At each time-point (1, 3, 5, and 7 days), samples were washed with PBS and the MTS reagent in serum- and phenol red-free DMEM was added into each well before incubating in the dark at 37 °C for 3 h. The absorbance values of supernatants were measured at 490 nm and a blank correction was performed. All experiments were done in quadruplicate.

### 2.11. Statistical Analysis

The obtained data was analyzed by one-way ANOVA with Bonferroni post-hoc test at *p* ≤ 0.05. Statistical analysis was performed using IBM SPSS statistics (Version 22, IBM, Armonk, NY, USA) software.

## 3. Results

### 3.1. Thiolation of SF

Primary amine on silk protein was attached by a sulfhydryl (SH) group by the thiolation reaction made from varying different 2-IT concentration (Figure 1A). The resulting SH groups of tSF solution were determined using the Ellman assay (Figure 1B). It was noticed that an increasing of 2-IT resulted in a greater amount of available SH groups. Our findings are in an agreement with previous reports that 2-IT introduces SH groups by attaching sulfanyl butanimidine groups to the primary amine on peptide chains [24].

### 3.2. Formation of Au-SF and Au-tSF Hydrogels

The appearances of regenerated SF and tSF (controls) and their mixtures with Au^3+^ incubated at 37 °C for 14 days are presented in Figure 2A. The regenerated SF and tSF solutions formed hydrogels within 7 days, which is in accordance with the increase of beta sheet content (Figure 2B,C). At 0.5 mM Au^3+^, the Au-tSF mixture (right vial; Figure 2A) formed a hydrogel within 1 day, while the gelation time was longer for the SF group (left vial; Figure 2A). At 1 mM Au^3+^, both SF and tSF hydrogels were formed in 1 day. Interestingly, distinct appearances of gels were noticed, i.e., the SF gel presented a red-to-purple color, while the tSF gels maintained their initial yellow color. For 5 mM Au^3+^, both SF and tSF formed hydrogels immediately after adding Au^3+^ into the solutions. However, due to rapid hydrogel formation, a local phase separation occurred, leading to non-homogeneous mixtures for both groups.

Beta sheet content was determined using FTIR followed by FSD and curve-fitting processes. The gelation of Au-SF mixtures using 0.5 mM Au^3+^ did not occur on day 1 as observed for the Au-tSF group, but their beta sheet content (28.7 ± 3.8% and 30.4 ± 1.5%, respectively) were not significantly different. Moreover, SF and tSF with 5 mM Au^3+^ formed hydrogel immediately after mixing, but the beta sheet content did not change as expected. Beta sheet content was only of 27.1 ± 3.3% in SF and 30.1 ± 3.5% in tSF, which were not significantly different from the values obtained for SF and tSF solutions without Au^3+^ (25.0 ± 1.4% and 27.3 ± 3.5%, respectively). The structural transition to beta sheet was faster in all Au^3+^-mediated groups compared to the samples without Au^3+^. These results imply that the SF and tSF structural transition is not the major mechanism of gelation induced by Au^3+^.

### 3.3. Formation of Dityrosine, Au-S Bonds, and AuNPs

The formation of dityrosine bonds can be examined by fluorescence spectroscopy using excitation and emission wavelengths of 320 and 410 nm, respectively. As shown in Figure 3A, the fluorescence intensity was proportional to Au^3+^ concentration. The fluorescence signals related to dityrosine formation rapidly increased when 0.5 mM Au^3+^ and 1 mM Au^3+^ were added into SF and tSF, respectively. Interestingly, lower concentration of Au^3+^ could induce dityrosine bonds in SF group, as demonstrated that of the SF group reached maximum intensity at 2 mM Au^3+^, whereas the tSF group showed maximum intensity at 3 mM Au^3+^.

Addition of Au^3+^ induced the formation of Au-S bonds and reduced the amount of free sulfhydryl groups (SH) on silk fibroin molecules (Figure 3B). Ellman reagent assay indicated a reduction of sulfhydryl groups to 13.6 µmol/g when 1 mM Au^3+^ was added into tSF solution. Due to the low amount of SH groups in SF solution, the SH group could not be quantified.

The formation of AuNPs was assessed from the UV–vis absorption spectra obtained from Au-SF and Au-tSF mixtures (Figure 3C,D). The appearance of the surface plasmon resonance band between 525 to 535 nm can be noticed, indicating the formation of AuNPs [25]. At 2 mM Au^3+^, formation of AuNPs could be detected in SF samples, while it did not occur in tSF group.

### 3.4. XPS Analysis of Au^3+^-Mediated SF and tSF Hydrogels

After the gelation of the mixtures composed of 1 mM Au^3+^ and 3% SF or 3% tSF, the samples were lyophilized and the XPS analyses were performed. XPS spectra in Au4f region of the samples are presented in Figure 4A,B. Three peaks were assigned within the Au4f_7/2_ region (BE 83.0–86.5 eV). The first and last peaks located near 84.0 and 85.5 eV represent the metallic Au^0^ and Au^3+^, respectively. The middle peak, which were shifted from the first peak around 0.5–0.6 eV, is related to Au^+^ species [22]. The quantification of each chemical state obtained from the calculated areas under peaks is presented in Table 1. The majority of the Au state in SF gel was Au^0^, while both Au^0^ and Au^+^ were noticed in the tSF gel at the same extent.

The analysis of free and bound thiol groups by XPS is shown in Figure 4C and Table 1. Due to the absence of thiol groups in SF, the spectrum in S2p region could not be obtained. The S2p spectrum of tSF was assigned into two components based on the spin-orbit splitting doublet, S2p_3/2_ (162.2 and 164.2 eV) and S2p_1/2_ (163.2 and 165.4 eV). The first component located at 162.2 eV was regarded as bound (chemisorbed) thiol group, while the peak at 164.2 eV corresponded to the free or weakly bound (physisorbed) sulfur [23].

### 3.5. SEM Analysis

The freeze-dried 1 mM Au-3% SF and 1 mM Au-3% tSF hydrogels were sectioned and their microstructures were analyzed by SEM (Figure 5). The Au-SF gel exhibited a leaf-like morphology and interconnected pores with size larger than 50 µm (Figure 5A). At a high magnification, distributed particles, with approximately 20 nm diameter, over the gel matrices were observed (Figure 5B). This could be an evidence of the formation of AuNPs. Different microstructures were noticed for tSF samples. The Au-tSF gel showed a micro- or nano-fibrous structure with smaller interconnected pores (Figure 5C), and without the presence of particles (Figure 5D).

### 3.6. Viscoelastic Properties of Au^3+^-SF and tSF Gels

The complex (G*), storage (G’) and viscous (G”) moduli over a range of frequency from 0.01 to 100 Hz of the samples are shown in Figure 6. Regenerated SF and tSF solutions were used as controls. For all samples, the moduli were constant over the frequency from 0.01 to 10 Hz before steeply rising until 100 Hz. G’ and G” of regenerated SF and tSF were almost equal which corresponded to their solution state. The addition of 0.5 mM Au^3+^ increased G* in the range of 1–20 Pa for SF samples, while the higher G* values (15–50 Pa) were obtained in tSF group. An increasing of Au^3+^ concentration resulted in gradually higher moduli. However, the moduli of Au-tSF samples were similar regardless of Au^3+^ concentration. Presumably, Au^3+^ at the concentrations greater than 0.5 mM did not significantly affect the viscoelastic properties of the obtained gels.

### 3.7. Cytocompatibility Evaluation of Au^3+^-SF and tSF Hydrogels

L929 fibroblasts were cultured on SF and tSF hydrogels prepared with different Au^3+^ concentrations (0, 0.5, 1, and 5 mM). Cells cultured on tissue culture plates (TCPs) were used as references. The metabolic activity of cells cultured on Au-SF and Au-tSF hydrogels in all Au^3+^ concentrations increased over the time course of 7 days culture period (Figure 7). However, a significant low metabolic activity at day 7 was noticed for both SF and tSF samples. Interestingly, without an addition of Au^3+^, a decrease of metabolic activity was observed in SF group, whereas tSF group displayed the higher metabolic activity. Despite metabolic activities were not as high as the ones obtained from the control TCP, SF, and tSF hydrogels with Au^3+^ supported the proliferation of L929, indicating the cytocompatibility of these hydrogels.

## 4. Discussion

Different strategies to induce the SF gelation have been proposed, namely crosslinking agents and techniques that stimulate the self-assembly process. In this work, Au^3+^, a metallic ion, was used as a promoter agent to induce SF and tSF gelation. The gelation mechanisms and the cytocompatibility of the resulting hydrogels were investigated.

The formation of 1 mM Au^3+^-3% SF hydrogels was accompanied with an obvious color change (from yellow to purple red), which can be related to the AuNPs formation (Figure 2A). The presence of AuNPs was confirmed by the surface plasmon resonance band at 525 nm which increased with higher concentrations of Au^3+^ (Figure 3C,D). The XPS results also indicated that Au^0^ is the predominant state (Figure 4) and SEM images showed distributed nanoparticles over the hydrogel matrix (Figure 5B). The presence of dityrosine bonds was studied by spectrofluorometry and the emission intensity increased with higher amounts of Au^3+^ added (Figure 3A). It can be proposed that, after the addition of HAuCl_4_, amino (-NH_2_) groups could act as reducing agents converting Au^3+^ to Au^+^. Afterwards, a reduction to Au occurs by accepting an electron from the tyrosine side chain. Hence, two tyrosyl radicals form a covalent dityrosine [12,14].

In the case of regenerated SF, the spontaneous gelation of SF is related to its self-assembly characteristic [3]. A less ordered random coil structure present in the solution evolves into a highly stable beta sheet conformation due to hydrophobic interactions and hydrogen bonding between repetitive peptide sequences. The increase in beta sheet structures results in stronger interactions of the peptide chains, leading to the gel formation.

The changes of the secondary structures from random coil to beta sheet did not support the kinetics of Au-mediated SF gelation. The presence of Au^3+^ cannot directly induce the development of protein secondary structures, which did not correspond with gel formation (Figure 2B,C). However, a more rapid increase of beta sheet conformation was observed in the presence of Au^3+^. It can be speculated that both the generation of protons due to the reduction of Au^3+^ and the dityrosine formation may interfere with the local pH. The pH, therefore, shifted towards the isoelectric point of SF (4.59–5.06), leading to SF chain aggregation and subsequent gel formation as it was previously reported by Matsumoto et al. [3].

tSF was synthesized by a simple chemical reaction using 2-IT and the obtained functionalized protein was mixed with Au^3+^. The resulting hydrogels maintained the initial color of the solution, indicating the absence of AuNPs formation (Figure 2A). The addition of Au^3+^ resulted in a decrease of available thiol groups (Figure 3B), confirming the disulfide (S-S) and gold-sulfide (Au-S) bonding. The XPS analysis also showed a higher relative amount of Au^+^ (Figure 4 and Table 1) and the formation of nanoparticulate features was not noticeable in the SEM images (Figure 5D). These results are in agreement with Jung et al. [13] who reported that thiol-presenting molecules, namely cysteine, homocysteine, and glutathione, prevent Au reduction and NPs formation. Like Au-SF hydrogels, the conformational transition of Au-tSF did not occur along with the gelation but the transition to beta sheet conformation was accelerated by the presence of gold. The generation of protons from the oxidation reaction of -SH to S-S or Au-S is proposed, resulting in a decrease of local pH and hence the kinetics of tSF gelation are enhanced. The proposed mechanisms of Au^3+^-mediated SF and tSF gelation are depicted in Figure 8.

At the highest tested concentration of Au^3+^ (5 mM), an immediate gelation occurred after mixing. However, non-homogeneous gels with partial precipitation were obtained (Figure 2A). Furthermore, there was no noticeable color change, due to AuNPs formation. This could relate to the high H^+^ from HAuCl_4_, which decreased the pH of the mixtures, leading to the protonation of tyrosine. Since pK_a_ of tyrosine is approximately 10, the proton-coupled electron transfer to Au^+^ is more difficult when the phenolic group of tyrosine is protonated [12].

The thiolation of SF and the presence of Au^3+^ resulted in a faster gelation compared with the kinetics of the gelation of SF with Au^3+^, especially at low Au^3+^ amounts. As shown in Figure 6, the modulus of 0.5 mM Au^3+^-tSF is higher than those of SF. This could imply the immediate formation of Au-S bonding in Au-tSF hydrogels due to click chemistry [18]. While the formation of dityrosine bonds, which is proposed as a major mechanism of Au-SF hydrogels, required a longer time [12].

Cytocompatibility results of Au-SF and Au-tSF hydrogels were performed using L929 cell line as a model. Normal proliferation of cells was noticed for all samples (Figure 7), indicating that the cytocompatibility of hydrogels was maintained even the presence of Au^3+^. Indeed, SF is well-known for its biocompatibility [2], but tSF remained inadequately studied. Our study investigated the compatibility of tSF with the cultured cells, confirming its potential as a biomaterial. Schedle et al. [26] confirmed the cytotoxicity of metallic ions, including HAuCl_4_, and they reported the TC50 (50% toxicity levels) of Au^3+^ was 0.077 mM. However, our developed hydrogels containing higher Au^3+^ amounts showed no toxicity for L929. In addition, SF and tSF might possess a preventive effect against the cytotoxicity of the metallic ion by the side chains of amino acid residues which act as reducing, stabilizing, or chelating molecules [27].

## 5. Conclusions

Au^3+^ is herein proposed to induce the gelation of regenerated SF and tSF and their underlying mechanisms were proposed. The addition of Au^3+^ significantly reduced the gelation time of SF and tSF, but the different gel microstructures were noticed, indicating different gelation mechanisms. Dityrosine bonding and the formation of AuNPs were proposed for Au^3+^-mediated SF hydrogels with the purple-red color. The presence of thiol groups in tSF prevented the reduction of Au^+^ to Au because of the strong Au-S interaction. The conformational transition of SF and tSF from random coil to beta sheet is proposed as a downstream mechanism after the generation of protons from dityrosine formation (for SF gel) or S-S or Au-S bonding (for tSF gel) and subsequent reduction of the local pH, leading to chain aggregation.

Au^3+^-mediated SF and tSF hydrogels showed good cytocompatibility when L929 cells were cultured at their surfaces. Further studies will be conducted to characterize in detail the biological performance of the developed hydrogels.

## Figures and Tables

**Figure 1 biomolecules-10-00466-f001:**
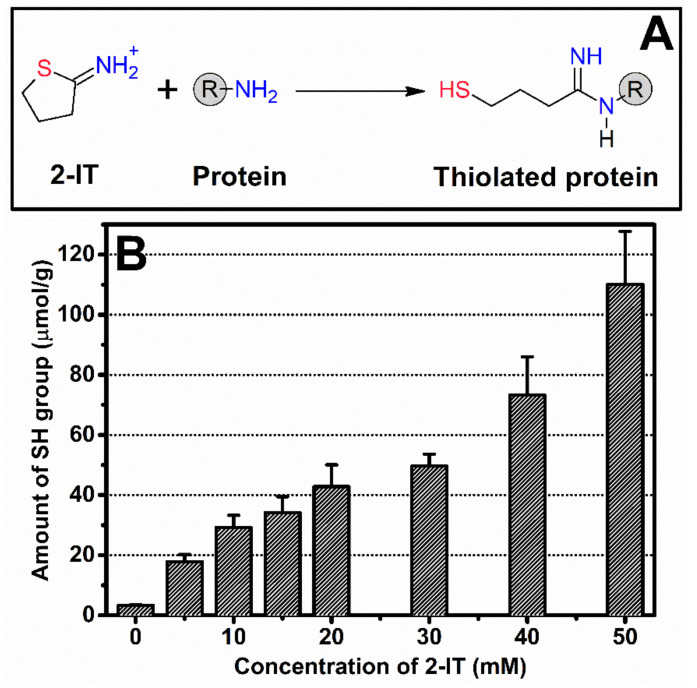
(**A**) Thiolation reaction of protein using 2-iminothiolane (2-IT) to attach a sulfanyl butanimidine group to a primary amine. (**B**) Amount of available sulfhydryl groups of thiolated silk fibroin (tSF) determined by the Ellman reagent assay as a function of 2-IT concentration.

**Figure 2 biomolecules-10-00466-f002:**
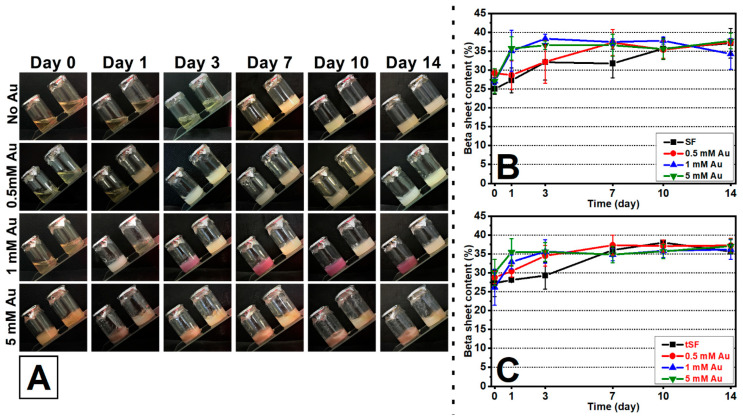
(**A**) Appearances of regenerated (no Au^3+^) and Au^3+^-mediated SF (left vial) and tSF (right vial) hydrogels incubated at 37 °C over 14 days. The amount of beta sheet determined from FTIR spectra using Fourier self-deconvolution (FSD) and curve-fitting techniques of the freeze-dried regenerated SF (**B**) and tSF (**C**) at different Au^3+^ concentrations.

**Figure 3 biomolecules-10-00466-f003:**
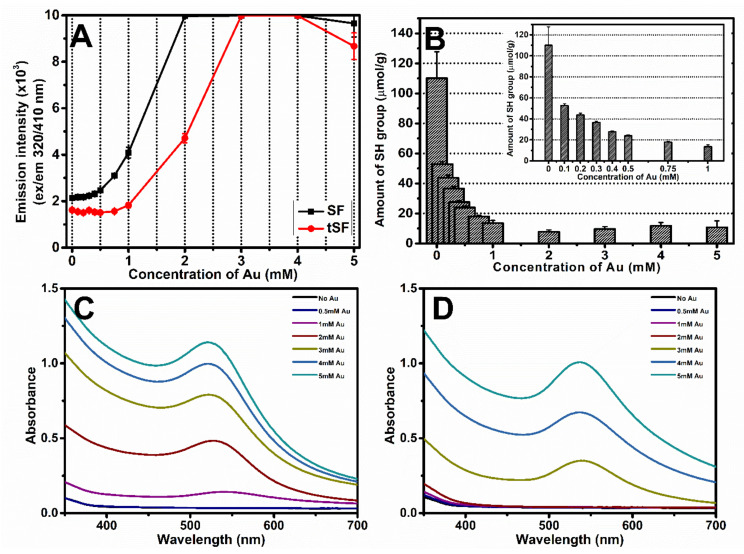
(**A**) Fluorescence measurements (excitation wavelength = 320 nm, emission wavelength = 410 nm) of SF and tSF, corresponding to the formation of dityrosine, after the addition of Au^3+^ at different concentrations, (**B**) Amount of available sulfhydryl (SH) groups in tSF in a presence of different amounts of Au^3+^. The inset shows the results for Au^3+^ concentrations below 1 mM. UV–vis absorbance spectra of the mixtures of (**C**) 1% SF and (**D**) 1% tSF at different Au^3+^ concentrations, presenting the surface plasmon resonance band of gold nanoparticles (AuNPs).

**Figure 4 biomolecules-10-00466-f004:**
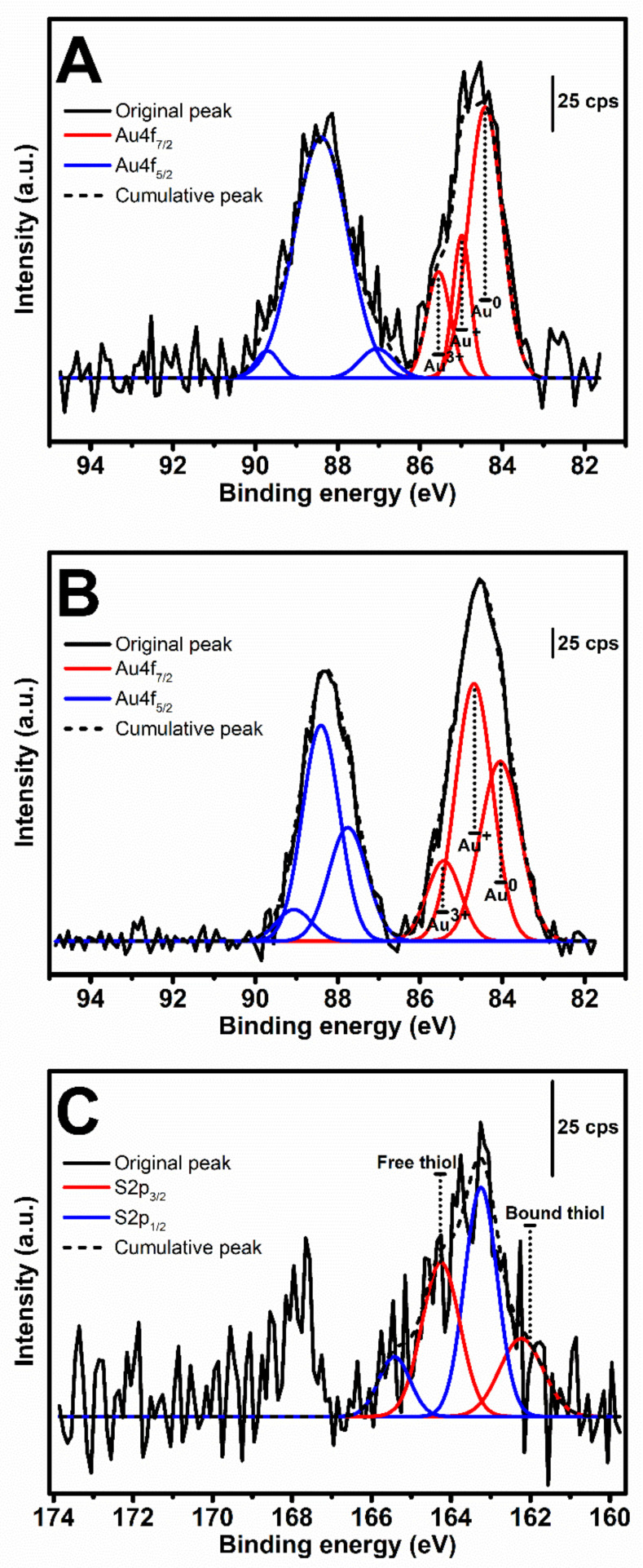
XPS spectra in Au4f region of (**A**) Au-SF and (**B**) Au-tSF, and in (**C**) S2p region of Au-tSF.

**Figure 5 biomolecules-10-00466-f005:**
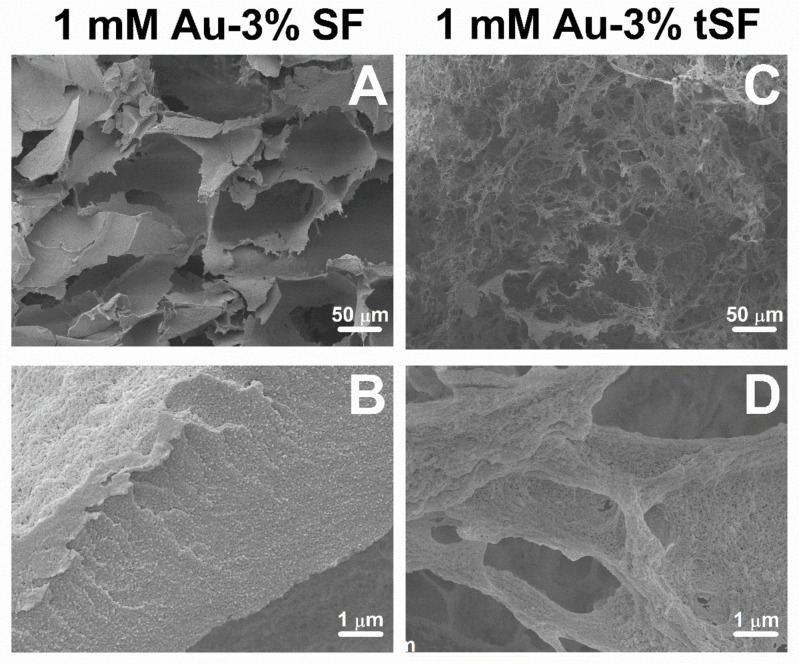
SEM images of the lyophilized 1 mM Au-3% SF (**A**,**B**) and 1 mM Au-3% tSF (**C**,**D**) hydrogels.

**Figure 6 biomolecules-10-00466-f006:**
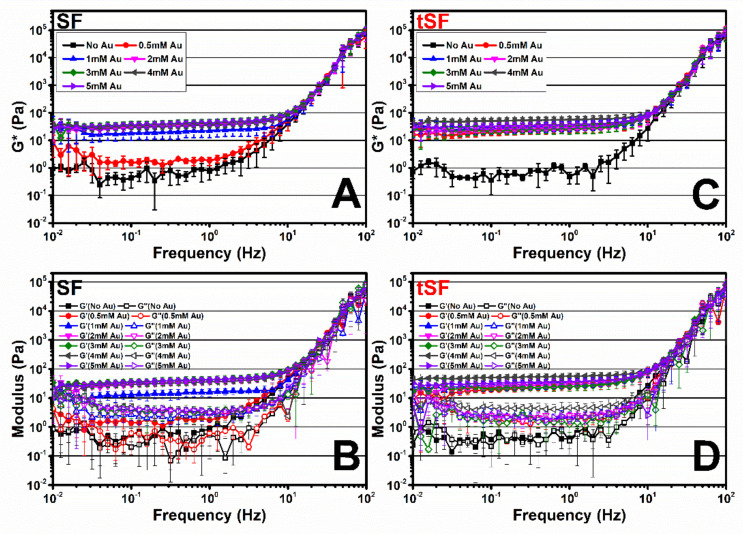
Frequency sweep experiments (0.01–100 Hz with 0.1% fixed strain, 37 °C) of regenerated SF and the mixtures of 3% SF (**A**,**B**) or tSF (**C**,**D**) with different Au^3+^ concentrations (1–5 mM). (**A**,**C**) Complex modulus (G*), (**B**,**D**) storage modulus (G’), and viscous modulus (G”) (*n* = 3).

**Figure 7 biomolecules-10-00466-f007:**
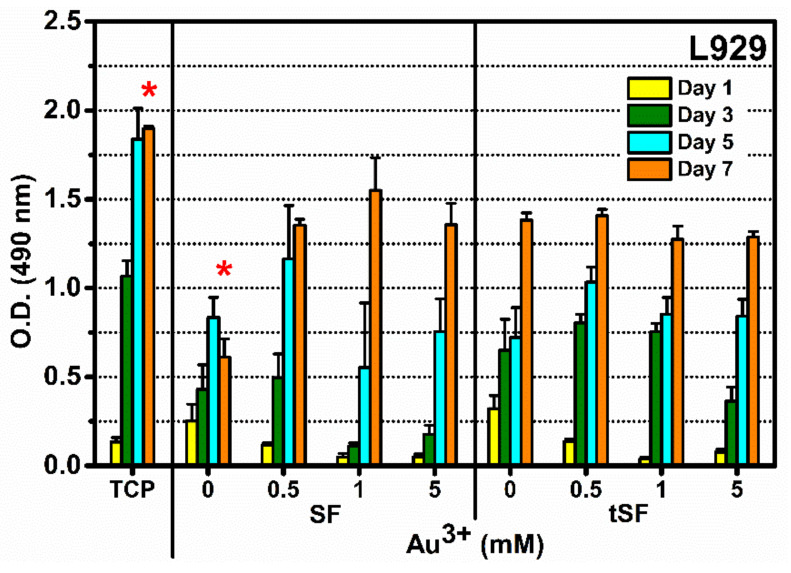
Cytocompatibility of SF and tSF hydrogels, and Au-SF and Au-tSF hydrogels with L929 rate fibroblasts. Cells were seeded on the hydrogels at 10,000 cells/cm^2^ and cultured for 7 days. Cell metabolic activity was assessed by MTS assay (*n* = 4) (TCP = tissue culture plate). * indicates statistical differences at *p* ≤ 0.05.

**Figure 8 biomolecules-10-00466-f008:**
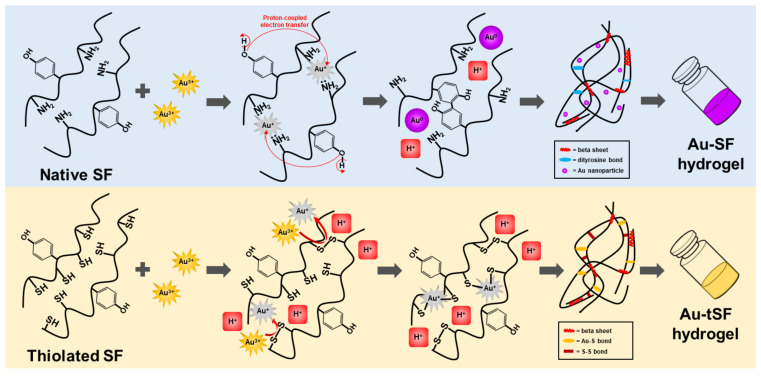
Proposed gelation mechanism of Au-SF and Au-tSF hydrogels. For SF, amino groups of protein side chains firstly coordinate with Au^3+^ and reduce to Au^+^. Tyrosine residues then donate electrons via proton-coupled transfer reaction to Au^+^ and reduce to Au, leading to dityrosine bonding and AuNPs formation. Protons (H^+^) are generated and induce SF beta sheet transition. For tSF, thiol groups donate electrons to Au^3+^, resulting in Au^+^, and form disulfide bonds. Subsequently, Au^+^ form bonds with available thiol groups without further reduction. The generation of H^+^ from S-S and Au-S bonds results in a beta sheet formation of tSF.

**Table 1 biomolecules-10-00466-t001:** Relative percentage of chemical composition determined from XPS spectra within Au4f_7/2_ and S2p region of the lyophilized 1 mM Au^3+^ with 3% SF or 3% tSF samples. Peaks areas of the different chemical states were calculated using Origin Pro 9.0 software.

XPS Region	Chemical State	1 mM Au + 3% SF		1 mM Au + 3% tSF	
		BE (eV)	Relative Amount (%)	BE (eV)	Relative Amount (%)
**Au4f_7/2_**	Au^0^	84.4	31.7	84.0	21.9
	Au^+^	84.9	8.9	84.6	28.9
	Au^3+^	85.5	8.8	85.4	8.3
**S2p_3/2_**	Bound S	N/A		162.2	18.2
	Free S			164.2	31.4

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
