# Peer review of "Exploring the Gelation Mechanisms and Cytocompatibility of Gold (III)-Mediated Regenerated and Thiolated Silk Fibroin Hydrogels"

_biomolecules, 2020, doi:10.3390/biom10030466_

Round 1

Reviewer 1 Report

Reviewer report to Authors

This is an interesting study in the field of hydrogel research.  This work reports the preparation of a silk fibroin  hydrogel systems using Au3+ salt as a chemical gelator. The gelation  behaviors and mechanisms of regenerated SF and thiolated SF (tSF) were compared. Au3+ mediated SF and tSF hydrogels showed different color appearances. The authors reported that the color of Au-SF hydrogels was purple-red, whereas the Au-tSF hydrogels maintained their initial solution color, indicating different gelation mechanisms. The cytocompatibility of the Au-SF and tSF hydrogels was demonstrated by culturing with a L929 cell line, indicating that the developed hydrogels can be promising 3D matrices for different biomedical applications. Both SF and thiol-functionalized SF (tSF) were selected as  base matrices to investigate and compared their gel formation in  the presence of Au3+. Furthermore, authors evaluated the cytocompatibility of the obtained hydrogels to assess their potential for biomedical applications  .

The main content of this paper is clear. Suitable experiments were conducted for the study and the results are reasonable to the hypothesis of the article .  But the manuscript need to  rewrite with more discussions and explanations for each experimental results to improve the quality  of the article. Then the manuscript  can be accepted for publication with major revision.

Suggested minor revisions are following.

  1. Page number 4 line 90 , it will be better to rewrite as amino group containing residues
  1. Please check 4°C and 37°C . Many places its not written in correct (Page number 5 line 114 and 116 ,119 ,123, 132 ,137,190,204,209,214,234,248)

  1. SEM images of the lyophilized Au-SF (A & B) and Au-tSF (C & D) hydrogels . Please specify the used concentration in the image title too.

  1. Any cell attachment/adhesion  studies conducted? Why L929 cell line used ? what is the application?

  1. The manuscript need to carefully revise and rewrite

Author Response

Responses to Reviewer 1

This is an interesting study in the field of hydrogel research.  This work reports the preparation of a silk fibroin hydrogel systems using Au3+ salt as a chemical gelator. The gelation behaviors and mechanisms of regenerated SF and thiolated SF (tSF) were compared. Au3+ mediated SF and tSF hydrogels showed different color appearances. The authors reported that the color of Au-SF hydrogels was purple-red, whereas the Au-tSF hydrogels maintained their initial solution color, indicating different gelation mechanisms. The cytocompatibility of the Au-SF and tSF hydrogels was demonstrated by culturing with a L929 cell line, indicating that the developed hydrogels can be promising 3D matrices for different biomedical applications. Both SF and thiol-functionalized SF (tSF) were selected as base matrices to investigate and compared their gel formation in the presence of Au3+. Furthermore, authors evaluated the cytocompatibility of the obtained hydrogels to assess their potential for biomedical applications.

The main content of this paper is clear. Suitable experiments were conducted for the study and the results are reasonable to the hypothesis of the article.  But the manuscript needs to rewrite with more discussions and explanations for each experimental result to improve the quality of the article. Then the manuscript can be accepted for publication with major revision.

 Suggested minor revisions are following:

  1. Page number 4 line 90, it will be better to rewrite as amino group containing residues

Ans     Thank you for your comments. The correction was made as suggested (Page number 4, Line 91).

  1. Please check 4°C and 37°C. Many places its not written in correct (Page number 5 line 114 and 116 ,119 ,123, 132, 137,190,204,209,214,234,248)

Ans     Celsius degree symbol was corrected following the guideline provided by NIST (https://www.nist.gov/pml/weights-and-measures/si-units-temperature) (Page number 5 Line 112, 115, 117,120, 124, 133, 138, Page number 7 Line 192, 206, 211, 215, Page number 8 Line 240, Page number 10 Line 255).

  1. SEM images of the lyophilized Au-SF (A & B) and Au-tSF (C & D) hydrogels. Please specify the used concentration in the image title too.

Ans     The concentration of Au, SF and tSF were specified in the figure caption of Figure 5 as suggested (Figure 5 was changed) (Page number 16 Line 337 and 338).

  1. Any cell attachment/adhesion studies conducted? Why L929 cell line used? what is the application?

Ans     We did not perform cell attachment/adhesion studies, since our aim was to evaluate the cytocompatibility of the developed hydrogels. Therefore, the cell metabolic activity assay (MTS assay) was performed to demonstrate that Au-SF and Au-tSF hydrogels were not cytotoxic.

          L929 cell line was selected for the biological evaluation since it is recommended by ISO as the standard cell line for biocompatibility tests of medical devices.  Furthermore, our results were able to be compared and discussed with others presenting in the literature (e.g. Schedle, A.; Samorapoompichit, P.; Rausch-Fan, X.H.; Franz, A.; Füreder, W.; Sperr, W.R.; Sperr, W.; Ellinger, A.; Slavicek, R.; Boltz-Nitulescu, G., et al. Response of L-929 Fibroblasts, Human Gingival Fibroblasts, and Human Tissue Mast Cells to Various Metal Cations. Journal of Dental Research 1995, 74, 1513-1520, doi:10.1177/00220345950740081301).

  1. The manuscript need to carefully revise and rewrite

Ans     We appreciate your comments. The manuscript has been revised by all co-authors.

Reviewer 2 Report

The manuscript is well written and the topic of article is consistent with the scope of the journal. The approach to introduce gold to accelerate the gelatin of silk fibroin solution and proposed mechanisms of gel formation are  interesting. I suggest considering a few comments:

  1. In the manuscript the Authors often describe the structure of compounds, but in the description of SEM observations word microstructure instead of structure should be used. Crystal structure tells us about the arrangement/distribution of atoms in the material (unit cell parameters, types of atoms, their orientation, type of bonds). Microstructure explains about the phases present, grain boundaries, grains, porosity, shape and size of the grains, distribution of precipitates etc. Please change.
  2. Figure 3. If it is possible, please change the scale in Fig.3 A, in order to show all recorded emission intensities (points at Au concentration equal 3 mM and 4mM are barely visible).
  3. It is known that dimerization of radicals in the case of tyrosine can give dityrosine or isodityrosine so dimmer formation can involve formation of a carbon–carbon bond or a carbon–oxygen bond. The covalent bond typically forms at the Cortho-Cortho position on the ring, although the Cmeta can also occur. (Miao GU, David C. Bode & John H. Viles. Scientific Reports volume 8, Article number: 16190 (2018)). Are these statements true in the case of Au-SF hydrogel? If yes, maybe the alternative mechanisms are also worth mentioning .
  1. Statistical analysis would be useful, especially in the case of in vitro experiments.

Author Response

Responses to Reviewer 2

The manuscript is well written and the topic of article is consistent with the scope of the journal. The approach to introduce gold to accelerate the gelatin of silk fibroin solution and proposed mechanisms of gel formation are interesting. I suggest considering a few comments:

  1. In the manuscript the Authors often describe the structure of compounds, but in the description of SEM observations word microstructure instead of structure should be used. Crystal structure tells us about the arrangement/distribution of atoms in the material (unit cell parameters, types of atoms, their orientation, type of bonds). Microstructure explains about the phases present, grain boundaries, grains, porosity, shape and size of the grains, distribution of precipitates etc. Please change.

Ans     We appreciate your comments. The word “structure” to describe the SEM result was changed as suggested (Page number 14 Line 327, 329, 331, Page number 19 Line 394).

  1. Figure 3. If it is possible, please change the scale in Fig.3 A, in order to show all recorded emission intensities (points at Au concentration equal 3 mM and 4mM are barely visible).

Ans     Unfortunately, the detection limit of the intensity obtained from fluorescence spectrometer we used is 10,000, while the fluorescence values of SF with 2-4 mM Au and tSF with 3-4 mM Au were higher than the detection limit.

  1. It is known that dimerization of radicals in the case of tyrosine can give dityrosine or isodityrosine so dimmer formation can involve formation of a carbon–carbon bond or a carbon–oxygen bond. The covalent bond typically forms at the Cortho-Corthoposition on the ring, although the Cmeta can also occur. (Miao GU, David C. Bode & John H. Viles. Scientific Reports volume 8, Article number: 16190 (2018)). Are these statements true in the case of Au-SF hydrogel? If yes, maybe the alternative mechanisms are also worth mentioning.

Ans     We sincerely thank your comment and your questions are very interesting. After reviewing several literatures, we have noticed that the formation of dityrosine can be the covalent bonds between C-C in either ortho or meta position and C-O to form isodityrosine. Unfortunately, we only performed fluorescence studies to evaluate the formation of dityrosine, while isodityrosine exhibits non-fluorescence absorption (Jason S. Jacob, David P. Cistola, Fong Fu Hsu, Samar Muzaffar, Dianne M. Mueller, Stanley L. Hazen, and Jay W. Heinecke. Human Phagocytes Employ the Myeloperoxidase-Hydrogen Peroxide System to Synthesize Dityrosine, Trityrosine, Pulcherosine, and Isodityrosine by a Tyrosyl Radical-dependent Pathway. J. Biol. Chem. 1996 271: 19950-. doi:10.1074/jbc.271.33.19950). Hence, it is unlikely to discuss the formation of isodityrosine considering our existing information unless other experiments are performed such as chromatographic analysis. However, the chromatography of macromolecules, such as SF or tSF, could be difficult to detect these phenomena.

  1. Statistical analysis would be useful, especially in the case of in vitro experiments.

Ans     The statistical analysis information was added in the Materials and Methods section. The Figure 7 was amended, and the statistic results were described in the Results section (Page number 17 Line 365-367, Page number 18 Figure 7 Line 378-379).

Reviewer 3 Report

The paper by Neves and colleagues focuses on mechanisms of gel formation induced by Au salts when included in solutions of regenerated silk fibroin (SF) or thiolated silk fibroin (tSF). The results convincingly demonstrate rapid gelation with the concomitant formation of Au nanoparticles and dityrosine crosslinks in the first case, as demonstrated by electron microscopy, dityrosine fluorescence, XPS measurements and plasmon resonance of gold nanoparticles. In the case of thiolated fibroin, the proposed mechanism involves formation of disulphide bonds, while no gold nanoparticles are observed. Both SF and tSF induced hydrogels show cytocompatibilty towards an L929 fibroblast cell line.

The paper proposes an interesting approach towards rapid gelation of fibroin induced by gold salts and it would certainly be valuable to the readership of “Biomolecules”.

I only have a question regarding the gelation mechanism of tSF : in vue of the proposed mechanism,  how a reducing agent (ie citrate, ascorbate and/or disulphide bond reducing agent) would influence the process ? Would it induce at least partial reversibility and / or weakening of the gels, with concomitant formation of gold nanoparticles ?

Author Response

Response to Reviewer 3

The paper by Neves and colleagues focuses on mechanisms of gel formation induced by Au salts when included in solutions of regenerated silk fibroin (SF) or thiolated silk fibroin (tSF). The results convincingly demonstrate rapid gelation with the concomitant formation of Au nanoparticles and dityrosine crosslinks in the first case, as demonstrated by electron microscopy, dityrosine fluorescence, XPS measurements and plasmon resonance of gold nanoparticles. In the case of thiolated fibroin, the proposed mechanism involves formation of disulphide bonds, while no gold nanoparticles are observed. Both SF and tSF induced hydrogels show cytocompatibilty towards an L929 fibroblast cell line.

The paper proposes an interesting approach towards rapid gelation of fibroin induced by gold salts, and it would certainly be valuable to the readership of “Biomolecules”.

I only have a question regarding the gelation mechanism of tSF: in vue of the proposed mechanism, how a reducing agent (ie citrate, ascorbate and/or disulphide bond reducing agent) would influence the process? Would it induce at least partial reversibility and/or weakening of the gels, with concomitant formation of gold nanoparticles?

Ans     Thank you for your interesting question. We have not investigated the effects from mixing reducing agents with Au-SF or Au-tSF mixtures. However, the Au nanoparticles, which present Au0, have been tried mixing with the SF or tSF solution but the gelation did not occur. It means that the redox status of Au is necessary for inducing the gelation process. As mentioned in Discussion section, the reduction of Au3+ is necessary for dityrosine formation in SF and disulfide bridging in tSF.

Furthermore, the FTIR results revealed an increase of the beta sheet upon the gelation process and this conformation is thermodynamically stable. It is quite difficult to reverse the gelation process unless the chemicals, which can destabilize the hydrogen bonds in the beta sheet structure, are applied.